# Head Pitch Angular Velocity Discriminates (Sub-)Acute Neck Pain Patients and Controls Assessed with the DidRen Laser Test

**DOI:** 10.3390/s22072805

**Published:** 2022-04-06

**Authors:** Renaud Hage, Fabien Buisseret, Martin Houry, Frédéric Dierick

**Affiliations:** 1CeREF Technique, Chaussée de Binche 159, 7000 Mons, Belgium; buisseretf@helha.be (F.B.); frederic.dierick@gmail.com (F.D.); 2Traitement Formation Thérapie Manuelle (TFTM), Private Physiotherapy/Manual Therapy Center, Avenue des Cerisiers 211A, 1200 Brussels, Belgium; 3Faculté des Sciences de la Motricité, UCLouvain, Place Pierre de Coubertin 1, 1348 Ottignies-Louvain-la-Neuve, Belgium; 4Service de Physique Nucléaire et Subnucléaire, UMONS, Research Institute for Complex Systems, Place du Parc 20, 7000 Mons, Belgium; 5Centre de Recherche FoRS, Haute-Ecole de Namur-Liège-Luxembourg (Henallux), Rue Victor Libert 36H, 6900 Marche-en-Famenne, Belgium; martin.houry@henallux.be; 6Laboratoire d’Analyse du Mouvement et de la Posture (LAMP), Centre National de Rééducation Fonctionnelle et de Réadaptation–Rehazenter, Rue André Vésale 1, 2674 Luxembourg, Luxembourg

**Keywords:** artificial intelligence, supervised machine learning, kinematics, head rotation test, neck pain

## Abstract

Understanding neck pain is an important societal issue. Kinematic data from sensors may help to gain insight into the pathophysiological mechanisms associated with neck pain through a quantitative sensorimotor assessment of one patient. The objective of this study was to evaluate the potential usefulness of artificial intelligence with several machine learning (ML) algorithms in assessing neck sensorimotor performance. Angular velocity and acceleration measured by an inertial sensor placed on the forehead during the DidRen laser test in thirty-eight acute and subacute non-specific neck pain (ANSP) patients were compared to forty-two healthy control participants (HCP). Seven supervised ML algorithms were chosen for the predictions. The most informative kinematic features were computed using Sequential Feature Selection methods. The best performing algorithm is the Linear Support Vector Machine with an accuracy of 82% and Area Under Curve of 84%. The best discriminative kinematic feature between ANSP patients and HCP is the first quartile of head pitch angular velocity. This study has shown that supervised ML algorithms could be used to classify ANSP patients and identify discriminatory kinematic features potentially useful for clinicians in the assessment and monitoring of the neck sensorimotor performance in ANSP patients.

## 1. Introduction

Understanding neck pain is an important societal issue [1,2]. The overall prevalence of neck pain in the general population ranges from 0.4% to 86.8% and is higher in women than in men [3]. It ranks fourth in terms of years lived with a disability [1,2]. The majority of patients with neck pain are now classified as experiencing a “non-specific” neck disorder [4,5,6], meaning neck pain that occurs without trauma, signs or symptoms of major structural pathology, neurologic signs or specific pathology [4]. Acute or subacute non-specific neck pain (ANSP) means that the pain has been present for less than three months [4,7]. The assessment of sensorimotor function, a generic term for tests that encompass all afferent and efferent information flows and central integration mechanisms that contribute to joint stability [8], has demonstrated its importance for a better understanding of the pathophysiological mechanisms associated with chronic neck pain [9]. Indeed, the assessment of sensorimotor function, especially through kinematics of the head rotations, seems promising for the identification of chronic neck pain [10] but also of acute-subacute neck pain as shown by our previous results, which suggested that sensorimotor changes may also occur rapidly after pain resolution [11]. Nevertheless, identification based on sensorimotor evaluation requires the ability to know what would characterize neck pain in terms of the kinematic features of movement. Sensorimotor assessment of neck motion based not only on position degrees of freedom but also on velocity and acceleration features (e.g., peak and average velocity) appears promising because it has high sensitivity and specificity [10,12].

Identifying kinematic features from time series and comparing them between groups, e.g., to evaluate treatments or classify neck pain motion across ageing, is a widely used method [11,12,13,14,15]. Here, we focus on a peculiar test called DidRen laser test, designed to assess sensorimotor control of the neck and about which the interested reader will find detailed information in [11,15,16]. The DidRen laser test consists of a standardized task in which yaw rotations of the head are performed from “target to target” in the same sequence. These are fast, accurate, and small-amplitude rotations (±30°) of the head in response to real visual targets to be hit by a laser beam placed on the subject’s head [17]. However, such a methodology removes a substantial amount of information from the raw time series. The DidRen laser test did not cause pain in the patients (probably because of the too low amplitude of the rotation < 30°). Since the relationship between pain and sensorimotor control is well-established [18,19,20,21], if the test had caused pain when performed, it could have increased the kinematic difference between ANSP patients and healthy subjects.

Resorting to artificial intelligence (AI) techniques may lead to another type of analysis, i.e., “the machine” should find the relevant specifications of time series. The present study is devoted to the latter case. AI is defined as a field of science and engineering concerned with the computational understanding of what is commonly referred to as intelligent behavior and the creation of artefacts that exhibit such behavior [22]. Machine learning (ML) is defined as a subfield of AI as follows: “Machine learning is a branch of artificial intelligence that systematically applies algorithms to synthesize the underlying relationships among data and information” [23]. ML provides an experiential “learning” that can be related to human intelligence as ML can improve its analyses by using computer algorithms. There are two main forms of ML: supervised and unsupervised [24]. In supervised ML (SML), the algorithms are provided with training data that are analyzed for the features that are important for classification and labelled. The model is then “trained” on these data before being tested on unlabeled data. In our case, the data will be measured in head rotations. In SML, data must first be labelled by a clinician (painful or not, for example) so that the model can learn to interpret them through pattern recognition. Then, the model is tested with unlabeled data to obtain an interpretation result [25,26]. Several algorithms can be trained for pattern recognition, such as logistic regression, support vector machine, decision tree, random forest, naïve Bayes or K-nearest neighbor [24]. Patterns may be representative of various features, among which pathology and pain, see e.g., [27].

The first aim of this work was to evaluate the discriminative ability of AI and SML methods in sensorimotor assessment of yaw angular displacement of the head in patients with ANSP compared with healthy control participants (HCP) with data from a previous study [11] obtained during the DidRen laser test [15,16,17]. A second aim of this work was to illustrate the potential of SML for clinicians in musculoskeletal physiotherapy [28]. In ecological situations, neck kinematics should be quickly assessed by a therapist using thresholds designed to identify relevant impairments in the history of patients with neck pain. We test whether SML can provide such kinematic values and therefore has predictive value for ANSP.

## 2. Materials and Methods

### 2.1. Patients and Participants

This study included 80 subjects (38 ANSP patients and 42 HCP) from a previous study [11]. Data were collected from February to December 2019. ANSP patients diagnosed by general practitioners were recruited from a consecutive sample in a private manual therapy center in Brussels, Belgium. Inclusion criteria for ANSP patients were acute-subacute (<3 months) non-specific neck pain with a Neck Disability Index (NDI) ≥ 8% [29] and a Numeric Pain Rating Scale (NPRS) > 3 [30,31,32,33,34]. HCP were recruited by one of the authors (RH) from a sample of convenience from colleagues at the university hospital and from acquaintances. They were included if they reported no neck symptoms: NDI < 8% [29], NPRS = 0 [30], and no pain on active head rotation and/or manual spinal assessment [35]. Characteristics of the ANSP patients and HCP are listed in Table 1. All subjects signed an informed consent form. The study was approved by the Academic Bioethics Committee (https://www.a-e-c.eu, (accessed on 30 January 2019) Brussels, B200-2018-103) and conducted in accordance with the Declaration of Helsinki. The authors confirm that all ongoing and related trials for this drug/intervention are registered (ClinicalTrials.gov: 04407637).

### 2.2. Protocol

The protocol was described in a previous study [11]. It essentially involved assessment of fast neck yaw rotations with the DidRen laser test [15,16] for ANSP patients and HCP, completed by manual examination of the painful spinal region for segmental tenderness. For ANSP patients, the manual examination served to confirm familiar pain and guide the treatment. For HCP, thanks to its high sensitivity (92%), the manual examination was used to exclude HCP if they had pain at one or more levels of the cervical spine and confirm that they are not healthy in the neck [35]. The DidRen laser test was used to standardize the rotational yaw movements of the participant’s head. Briefly, participants wore a helmet to which a laser was attached. They pointed the laser as fast as possible at three targets equipped with photosensitive sensors (Figure 1A,B). The angular separation of targets is 30°, and the sequence was fixed: center-left-center-right-center. Participants were asked to perform the sequence as fast as possible.

During the DidRen laser test, head angular displacement kinematics were recorded in 3D (yaw, pitch, and roll) using the DYSKIMOT inertial sensor [36]. The detailed description of the sensor can be found in the study by Hage et al. [36]. The sensor consists of a 3-axis accelerometer, a gyroscope and magnetometer, and a temperature sensor. These internal components respectively measure acceleration (in g, ±16 g), angular velocity (in °/s, ±2000°/s), and magnetic field (in gauss, ±16 gauss). The sensor recorded the motion at a sampling frequency of 100 Hz. The DYSKIMOT sensor was placed in front of the helmet (Figure 1C), with the yaw-axis (or X) in the vertical direction. The pitch-axis (or Y) was aligned with subject’s medio-lateral axis at the start of the test and the roll-axis (or Z) was aligned with the antero-posterior axis. The head rotation demanded in the DidRen laser test is oriented along the yaw-axis. Note that the subjects were not instructed to realize pitch or roll rotations of the head during the test.

### 2.3. Data Analysis

#### 2.3.1. Dataset and Pre-Processing

In our previous papers [11,15], we analyzed the same dataset by resorting to “standard” statistical tests: we calculated several kinematical features of the angular position, speed, and acceleration time series (e.g., peak speed, time to reach peak, etc.). Then we showed that some parameters were significantly different between ANSP and HCP [11], and that age also had a significant impact on the parameters [15]. In the present study, we re-analyse the same dataset by using the raw sensor data to train various ML algorithms with the goal of finding an algorithm able to separate ANSP and HCP. To our knowledge, it is the first time that such ML techniques are used in the field of neck pain. The dataset consists of 7 time series for each participant: time, angular velocity (three components labelled GyrX, GyrY, GyrZ), and acceleration (AccX, AccY, AccZ). Then, a pre-processing procedure was applied to convert each time series into a summary format for all participants. Each time series is summarized with 7 statistical descriptors: 1st, 2nd, and 3rd quartiles, mean, minimum, maximum, and standard deviation. The result is a dataset with 186 inputs and 42 features (6 time series × 7 descriptors). Each set of statistical descriptors is labeled as ANSP (value 1) or HCP (value 0).

#### 2.3.2. ML Algorithms and Determination of the Best Performer

It is generally difficult to determine a priori which ML algorithm performs best on a given dataset [37]. Therefore, several algorithms were tested to determine the most appropriate for classifying ANSP patients and HCP: K-Nearest Neighbor (KNN), Linear Support Vector Machine (Linear SVM), Non-linear Support Vector Machine Radial Basis Function (SVM RBF), Decision Tree (DT), Random Forest (RF), Adaptive Boosting (AdaBoost), and Gaussian Naive Bayes (GaussianNB).

The comparison between selected algorithms was based on metrics such as accuracy and the Area Under Curve (AUC) score, computed from the Receiver Operating Characteristic (ROC) curve. These metrics are only meaningful if the predictions are based on data that the ML algorithms have never learned. Therefore, the dataset was randomly split into two parts. The first part is the “training set”, which consists of 80% of the dataset used to train the ML algorithms. The second part (remaining 20%) is the “test set” used to make the predictions with the trained ML algorithms. The validation of the ML algorithms is performed by n-fold cross-validation [38]. To minimize the biases associated with the training dataset, 100 different cross-validations were performed on mixed data for each selected ML algorithm. The hyperparameters of the ML algorithms were optimized using the Grid Search method [39] that finds the best combination of fixed hyperparameters based on n-fold cross-validation.

For KNN, the optimized parameters were the following: the number of neighbors (n_neighbors: 3, 5, 8, 10), the weighting function (weights: uniform, distance) and the algorithms used to compute the nearest neighbors (algorithms: Brute-Force (BF KNN or BF KNN), kd_tree, auto, ball_tree). For Linear SVM, different values for the regularization parameter or C-parameter (0.1, 1, 10, 100, 1000) were used in the evaluation to test the dependence of the approach on the C-parameter. For SVM RBF, the C-parameter (0.001, 0.01, 0.1, 1, 10, 100) and the kernel coefficient Gamma (0.001, 0.01, 0.1, 1, 10, 100) parameter were optimized. For DT, the optimized parameters were the maximum depth of the tree (max_depth: 1, 5, 10, 100), the function to measure the quality of the splits (criterion: gini, entropy), and the strategy to select the split nodes (splitter: best, random). For RF, the optimized parameters were the maximum depth of the tree (max_depth: 1, 5, 10, 100), the number of trees in the forest (n_estimators: 1, 5, 10, 100), and the number of features considered in the search for the best split (max_features: 1, 5, 10, 100). For Adaboost, the optimized parameters were the maximum number of estimators at which boosting stops (n_estimators: 1, 5, 10, 50, 100, 500) and the weight applied to each classifier at each boosting iteration (learning_rate: 0.000001, 0.001, 0.1, 1, 5, 10, 100). For GaussianNB, the optimized parameter was the ratio of the largest variance of all features added to the variances for computational stability (var_smoothing: 0.0000001, 0.01, 1, 10, 100).

All the computations related to the determination of the best performer were made in Python 3.8 and SciKit-Learn 1.0.2 software.

#### 2.3.3. Determination of Most Informative Kinematic Features and Logistic Regressions

The most informative kinematic features, i.e., the features that trigger the most predictions, were computed by using the Sequential Feature Selector (SFS) forward and backward [40]. The backward SFS removes the poorest features one by one, while the forward SFS identifies the best combination of features. In both cases, the result is a list of kinematic features that performed best according to the AUC score. Each SFS was run 700 times (7 ML algorithms × 100 random data repartitions). Once the most informative kinematic feature was identified, a logistic regression was performed by using it, and the accuracy of this logistic regression was computed. Another logistic regression on total DidRen laser test duration was also performed to compare the present results to the unique outcome of the original DidRen laser test [17].

All the computations related to the determination of the most informative features and ML algorithms were made in Python 3.8 and SciKit Learn 1.0.2 software.

## 3. Results

### 3.1. Optimal Hyperparameters and Performance Metrics of ML Algorithms

Optimal hyperparameters are presented in Table 2. Performance metrics of the selected ML algorithms are given in Table 3. The least performing ML algorithm is the KNN, and the best performing one is the linear SVM with an accuracy of 82% and AUC of 84%. We show in Figure 2 the ROC curve of the Linear SVM, which is the best ML algorithm we found to classify ANSP patients and HCP.

The ROC curve plots the False Positive Rate ($\frac{\mathrm{FP}}{\mathrm{FP}+\mathrm{TN}}$) and the True Positive Rate ($\frac{\mathrm{TP}}{\mathrm{TP}+\mathrm{FN}}$) at all thresholds of Linear SVM classification.

### 3.2. Most Discriminative Features and Logistic Regressions

The most discriminative feature, regardless of the ML algorithm and SFS, was the first quartile of head pitch angular velocity (or GyrY), which ranked first 813 times in 1400. The second most discriminative feature was the median of head pitch angular velocity (ranked first 444 times in 1400). Thus, the pitch angular velocity appears to be the best discriminating feature to differentiate ANSP patients and HCP assessed with the DidRen laser test.

A logistic regression based on the median of head pitch angular velocity led to an accuracy of 77%. A logistic regression based on total duration of the DidRen test led to an accuracy of 63%.

## 4. Discussion

Our findings showed the effectiveness of the kernel linear SVM classifier in distinguishing ANSP patients from HCP. The accuracy of the linear SVM was 82% and the AUC score was 84%. The interpretation of the AUC score should be evaluated in terms of the importance given to its accuracy. We can assume that the medical community in the field of oncology prefers an AUC score close to 100%. Considering, on the one hand, the musculoskeletal field and, on the other hand, in relation to the non-specific pathology, the comparison between ANSP patients and HCP, which shows a great variability of the results [41], an AUC score higher than 80% can be considered satisfactory. As mentioned in the Introduction, the DidRen laser test did not cause pain in ANSP patients. This feature may help ANSP patients to show kinematic features such as HCP, which may increase the number of false negatives. Therefore, a larger rotation amplitude than 30° may decrease the false-negative rate.

Seven time series (time and kinematic data) related to yaw, pitch, and roll angular displacement and velocity of the head, which can be easily acquired with a single inertial sensor, were used to train the selected ML algorithms. However, regardless of the ML algorithm and SFS, not all axes of head motion have good discriminative information, as the two best discriminating kinematic features were related to head pitch. The accuracy was best with the linear SVM and lowest with all other selected ML algorithms, such as the non-linear SVM (RBF). The same finding regarding the superiority of linear SVM over RBF has already been observed in a study with limited sample size (17 young and 17 old subjects) aimed at detecting age-related changes in running kinematics [42]. For use in future clinical trials with kinematic variables with limited sample size, linear SVM may thus be a suitable option.

Like other studies using ML algorithms to detect kinematic changes in healthy or pathological subjects [42,43,44,45], our study is based on a rather small dataset in terms of typical AI calculations, but the results are consistent with the conclusions of [46]. While conducting observational sensorimotor assessment studies with large datasets holds promise for improving the understanding and management of various pathologies, here, the pathophysiological mechanisms associated with neck pain, the use of small datasets may also allow for a reduction in selection bias [46]. In addition, it is worth noting that an SVM has already been used in the musculoskeletal field to compare temporomandibular patients with control subjects [47]. With a smaller sample (10 patients and 10 control subjects), they achieved an average predictive accuracy of 60% (*p* = 0.10) [47]. The linear SVM algorithm is affordable with today’s standard devices: a tablet computer could efficiently post-process the data from any wearable inertial sensor. Note also that a logistic regression based on head pitch angular velocity could be easily implemented on any smartphone, but with a lower accuracy of 77%.

The main discriminatory information used by the linear SVM algorithm to distinguish ANSP patients from HCP are the first quartile and the median of head pitch angular velocity. The two best kinematic discriminating features differed from those obtained by inferential statistical analysis, suggesting that ML approaches are complementary and clinically useful to detect kinematic changes in patients with ANSP.

HCP have larger medians and quartiles for head pitch angular velocity (computed from GyrY time series) than ANSP patients, making the *Y*-axis a highly discriminatory direction that should be prioritized for future clinical trials with the DidRen laser test. Our results may seem counterintuitive at first, because the DidRen laser test consists of a sensorimotor assessment organized around the *Z*-axis, i.e., during the execution of yaw rotations of the head. Thus, it would stand to reason that the GyrX time series should contain most of the discriminative information. Nevertheless, it is interesting to note that the sensorimotor disturbances in ANSP patients may be highlighted by the stronger secondary coupled motion during yaw rotations. There seems to be a reason for this, because biomechanically, coupled bending rotations in the cervical spine lead to a compensatory roll rotation, which compensates for the yaw rotation of the head, and the associated coupled movements observed during pitch head movements [48]. Indeed, in HCP, we can observe that yaw head rotation (55.5 ± 10.8°) is coupled with a larger pitch motion (16.3 ± 11.4°) than roll motion (4.6 ± 6.2°) [48]. If we apply these considerations to patient assessment, this information may be of clinical interest because 3D motion analysis may be a useful tool for assessing postural changes in the cervical spine during sitting, but also because altered kinematics are associated with decreased performance, e.g., neck velocity and neck motion fluidity in functional movement tasks, in people with neck pain [49].

The present discussion suggests that the ML algorithms can provide relevant functional variables and thus optimize the prediction of ANSP status during the DidRen laser test. To further illustrate this point, we mention that total test duration was the only parameter measured in the original version of the DidRen laser test [17]. Logistic regression performed with duration yields lower accuracy than that obtained with the median of pitch head rotation alone, the latter parameter being favored by linear SVM.

In experimental studies, control and experimental groups are usually formed in such a way that no significant difference is observed in parameters such as age, ethnicity, gender, and degeneration/maturation stage, except for the variable of interest. In our case, this means that ANSP patients and HCP groups should differ only in terms of NDI and NPRS. Age is also significantly different in our groups, but we do not believe this is problematic for our purpose. Indeed, ML algorithms are designed to distinguish between HCP and ANSP patients. To find out the characteristics of ANSP patients, it is logical to compare them with the “healthiest” subjects, i.e., our HCP group. On the other hand, a control group with too young subjects would also have led to bias, since we have shown in a previous study that the kinematic behaviors recorded with the DidRen laser test have a U-shaped or inverted U-shaped age profile, making the differences between young and old particularly clear [16]. Because the prevalence of degenerative joint changes increases with age [4,50], possibly leading to movement limitations (yaw rotation steadily decreases between the ages of 30 and 60) [51], we selected HCP using a very sensitive manual examination [35]. After this examination, positive control subjects (with potential neck disorders) were excluded, and because their average age was higher (see [11]: the mean age of the excluded control subjects was 43.3 years), the average age of our HCP group decreased compared with the ANSP group. It is worth noting that a significant age difference between control and disease subjects was already found in a study aimed at developing and determining the predictive performance of ML models to distinguish between different subtypes of low-back pain and healthy control subjects [52]. For this purpose, as we did, they did not include age as a predictor when constructing the model [52].

It has already been shown by authors of the present study that several kinematic features of head rotation movements were significantly different in HCP and ANSP patients in terms of statistical tests comparing means [11]. The novelty of our results can also be outlined by comparing them with similar studies. The next logical step in the kinematical analysis of head rotation movements is to investigate whether some kinematic parameters can be used as predictors of neck pain or not. Bahat et al. performed simple logistic regressions on all selected kinematical parameters and found that it was the case [10]. For example, they found a sensitivity of 91% (right) and 94% (left) and a specificity of 95% (right and left) for head peak yaw angular velocity, and the maximum AUC was obtained with head peak pitch angular velocity [10]. Note that the discriminative power of head peak yaw angular velocity was shown in [11,13], where a test in a virtual (VR) environment was performed. The extent to which our current results hold in a VR environment is an open problem that we leave for future work.

We reach the same conclusion as [10] regarding pitch angular velocity by using a more model-independent approach, i.e., by allowing machine learning algorithms to sort out the most discriminant features from the raw data. We also go beyond simple logistic regressions by including all relevant features in a single ML algorithm. Indeed, converting time series into scalar variables may remove a substantial amount of information contained in the original time series that could lead to extra false negative results or inaccurate predictions [53]. Roijezon et al. used linear discriminant analysis to identify neck-pain patients, i.e., the same kind of methodology as ours, but obtained lower sensitivity and specificity than in the present study: they found a sensitivity of 74.6% and a specificity of 73.5% for classification based on head peak yaw angular velocity [12]. Thus, to our knowledge, this is the first time that ML algorithms have been applied to the raw sensor data recorded during head rotation to find a multi-feature classification algorithm for identifying ANSP patients. A clear advantage of this type of algorithm, in addition to the high accuracy currently achieved, is that it can be systematically improved by increasing the size of the data set and allowing the algorithm to “learn” from the new data.

In summary, we have shown that AI can help identify patients suffering from neck pain using the DidRen laser test augmented by an inertial sensor. In our approach, the accuracy and AUC scores are computed from inertial sensor’s kinematic data. The obtained ML algorithm can be implemented in any tablet or smartphone and lead to an “augmented DidRen laser test”; hence, our results may be transferred to daily clinical practice. In our opinion, the best way to merge the DidRen laser test and an inertial sensor is to develop a VR version of the test: it will improve the standardization of the test through the standardization of the environment, and any VR device has at least one inertial sensor able to collect the needed data. Such a work is in progress, see e.g., [54]. Using AI to interpret sensor data can in principle be used in other movements than the rotation demanded in the DidRen laser test, but then the AI training must be made for each different motion, which outlines the necessity of defining standardized movements in clinical tests. Today, there is still no clinical gold standard for diagnosing acute neck pain, but the use of the DidRen laser test and AI appears to be a promising candidate to provide clinically useful information that can improve patient management. The diagnostic ability of our framework has been proven in the present study, but it is worth mentioning the possibility of data storage offered by sensor technology. The more data that will be stored, the more the ML accuracy will be refined, i.e., our diagnostic algorithm is systematically improvable over time. Moreover, the same test can be performed at various points of one patient’s treatment to assess his/her evolution. One last feature of our approach is that it identifies key kinematic parameters (such as peak angular speed) on which therapists can focus to follow one patient’s evolution.

## Figures and Tables

**Figure 1 sensors-22-02805-f001:**
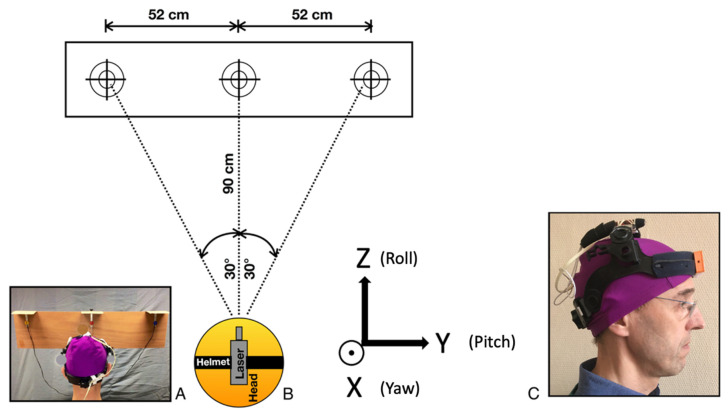
Description of the DidRen laser test. (**A**) Rear view of head position in front of the targets. (**B**) Schematic top view of the experimental setup with the three photosensitive sensors. The reference frame of the sensor is displayed when the head is in rest position. Coordinate system used in the study is also shown with the yaw (*X*-axis), pitch (*Y*-axis), and roll (*Z*-axis) rotations of the head during the test. (**C**) Helmet worn by an HCP (here RH) with laser on the top of the head and DYSKIMOT inertial sensor on the forehead.

**Figure 2 sensors-22-02805-f002:**
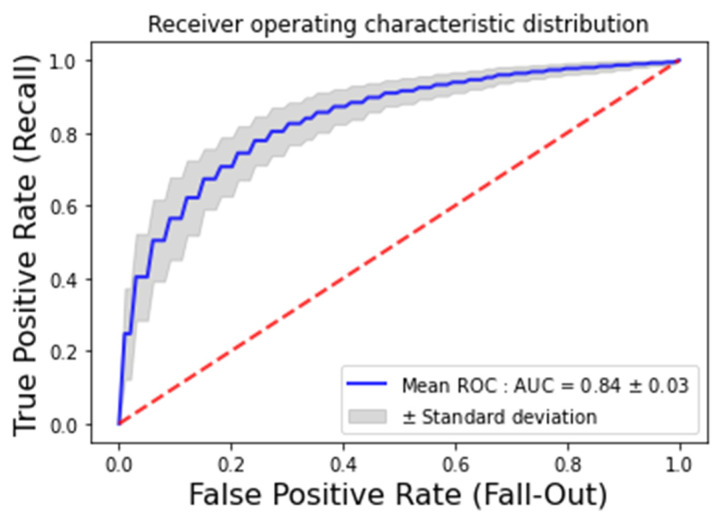
Receiver Operating Characteristic (ROC) curve of Linear SVM (in blue). The dotted red line represents the worst possible scenario, a random classifier.

**Table 1 sensors-22-02805-t001:** Characteristics of the acute and subacute non-specific neck pain (ANSP) patients and healthy control participants (HCP). *p*-values resulted from *t*-test for age and BMI, Mann–Whitney U-test for NDI and NPRS, and Chi-2 for gender.

	ANSP Patients (*n* = 38)	HCP (*n* = 42)	*p*-Values
Age (years), mean ± SD	46.2 ± 16.3	24.3 ± 6.8	<0.001
Gender *n* (men/women), (%)	21 (55%)/17 (45%)	27 (64%)/15 (36%)	0.55
BMI (kg m^−2^), mean ± SD	23.5 ± 3.2	21.5 ± 4.2	0.014
NDI (100), median [Q1–Q3]	22 [16–31.5]	0 [0–0]	<0.001
NPRS, median [Q1–Q3]	6 [4–7]	0 [0–0]	<0.001

BMI: body mass index, NDI: neck disability index, NPRS: numeric pain rating scale.

**Table 2 sensors-22-02805-t002:** Optimal hyperparameter values: Number of neighbors (n_neighbors), Regularization parameter (C-parameter), Kernel coefficient (gamma), maximum depth of the tree (max_depth), number of trees in the forest (n_estimators), and number of features to consider when looking for the best split (max_features).

ML Algorithm	Hyperparameters
BF KNN	n_neighbors = 5, weights = “distance”
Linear SVM	kernel = “linear”, C = 10
SVM RBF	gamma = 0.001, C = 100
DT	max_depth = 1, criterion = “entropy”, splitter = “best”
RF	max_depth = 10, n_estimators = 100, max_features = 10

BF KNN: Brute-Force K-Nearest Neighbors, SVM: Support Vector Machine, RBF: radial basis function, DT: Decision Tree, RF: Random Forest.

**Table 3 sensors-22-02805-t003:** Performance metrics of the selected ML algorithms.

ML Algorithm	Accuracy	AUC Score
BF KNN	0.66 ± 0.03	0.51 ± 0.07
Linear SVM	0.82 ± 0.03	0.84 ± 0.04
SVM RBF	0.65 ± 0.05	0.57 ± 0.09
DT	0.74 ± 0.03	0.70 ± 0.04
RF	0.76 ± 0.03	0.76 ± 0.04
AdaBoost	0.75 ± 0.04	0.76 ± 0.05
GaussianNB	0.77 ± 0.03	0.82 ± 0.03

BF KNN: Brute-Force K-Nearest Neighbors, SVM: Support Vector Machine, RBF: radial basis function, DT: Decision Tree, RF: Random Forest, AdaBoost: Adaptive Boosting, GaussianNB: Gaussian Naive Bayes, AUC: area under curve.

## Data Availability

Data are available at https://osf.io/rsuh2/ (accessed on 12 March 2022).

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
