# Peer review of "Head Pitch Angular Velocity Discriminates (Sub-)Acute Neck Pain Patients and Controls Assessed with the DidRen Laser Test"

_sensors, 2022, doi:10.3390/s22072805_

Round 1
Reviewer 1 Report
Authors present with clarity the processing of the results obtained from the Clinical Trial: 04407637.
The paper organization is coherent.
Also, they claim that previously they found found several differences between ANSP patients and HCP on several features using inferential statistical analysis.
Authors should provide more details regarding the improvements achieved in this work while compared against [15] and [11]. A table would be very useful.
Could you elaborate more on the false positives? Is it a method to capture them?
Reviewer 2 Report
The authors conducted secondary data analysis using different machine learning algorithms to classify acute or subacute neck pain and healthy participants, based on kinematic measures from a head-mounted inertia sensor during the DidRen laser test. Overall the paper was well presented with details on the methods, results, and discussions.
Here are my comments:
- The title is long and confusing. Linear support vector machines are methods, while head pitch angular velocity is a variable. In addition, based on the article’s content, there is not sufficient data to support they “are the best performers” to distinguish acute neck pain.
- In the introduction session, the importance of sensorimotor function assessment was discussed in chronic neck pain, but this article focused on acute or subacute neck pain. Will the same assessment be shared between chronic and acute neck pain?
- The clinical significance of the work can be further explained in the discussion. It is not clear what the current clinical gold standard to diagnose acute neck pain is. Was the DidRen test standardized clinical practice for acute neck pain? The results from this study were based on data from the DidRen test.
Round 2
Reviewer 1 Report
Authors have addressed the reviewers' comments.
Author Response
Thank you for your comment.
Reviewer 2 Report
Thank you for the added information, especially on the novelty and significance of the work. It is much clear to me now.
One more question: The accuracy and AUC score results were based only on kinematic data from the inertia sensor. How will this information integrate into the DidRen laser test? Or is it possible to use the inertia sensor with a series of head movements, combined with AI, to discriminate between the ANSP patients and healthy controls?
